# An Overview of Epigenetic Changes in the Parkinson’s Disease Brain

**DOI:** 10.3390/ijms25116168

**Published:** 2024-06-03

**Authors:** Anthony Klokkaris, Anna Migdalska-Richards

**Affiliations:** Department of Clinical and Biomedical Sciences, Faculty of Health and Life Sciences, University of Exeter Medical School, University of Exeter, Exeter EX2 5DW, UK; ak735@exeter.ac.uk

**Keywords:** epigenetics, Parkinson’s disease, DNA methylation, DNA hydroxymethylation, histone modifications, non-coding RNAs, microRNAs, epigenetic therapy

## Abstract

Parkinson’s disease is a progressive neurodegenerative disorder, predominantly of the motor system. Although some genetic components and cellular mechanisms of Parkinson’s have been identified, much is still unknown. In recent years, emerging evidence has indicated that non-DNA-sequence variation (in particular epigenetic mechanisms) is likely to play a crucial role in the development and progression of the disease. Here, we present an up-to-date overview of epigenetic processes including DNA methylation, DNA hydroxymethylation, histone modifications and non-coding RNAs implicated in the brain of those with Parkinson’s disease. We will also discuss the limitations of current epigenetic research in Parkinson’s disease, the advantages of simultaneously studying genetics and epigenetics, and putative novel epigenetic therapies.

## 1. Introduction

Parkinson’s disease (PD) is a debilitating movement disorder and the second-leading cause of neurodegeneration, affecting more than ten million individuals worldwide [1]. The prevalence of the disease increases sharply with age, with 1.7% affected by the age of 80–84 [2]. With a rapidly aging population, the number of people affected is only expected to increase. It is estimated that the number of individuals with PD will double between 2015 and 2040 [3]. PD already significantly contributes to the global burden of disease, with an economic burden of USD 51.9 billion per year in the United States alone [4].

Parkinson’s is associated with a number of motor symptoms, including resting tremor, bradykinesia, rigidity and postural instability [5]. In addition, the vast majority of individuals with PD also experience a range of non-motor complications, such as sleep disturbances, olfactory deficits, autonomic dysfunction, cognitive impairment and depression [5,6]. While some non-motor symptoms such as hyposmia, depression and REM sleep behavior disorder typically precede the onset of motor symptoms, other symptoms such as cognitive impairment, dementia, anhedonia and psychosis more commonly appear later [6,7]. Parkinson’s is also linked with substantial morbidity and mortality, with pneumonia and cardiovascular complications accounting for the majority of deaths [8].

Pathologically, PD is characterized by the progressive loss of dopaminergic neurons in the substantia nigra pars compacta (Figure 1). These neurons project to the striatum, and consequently, striatal dopamine levels are typically depleted in diseased individuals, accounting for many of the motor defects [9]. Dopaminergic neuron loss is normally also accompanied by the formation of α-synuclein aggregates in intracellular Lewy body inclusions, the pathological hallmark of PD [9]. Increased levels of α-synuclein may be toxic to human neurons and are believed to lead to further dopaminergic neuron loss [10]. It has also been suggested that misfolded α-synuclein may propagate along anatomically connected axons in a prion-like manner, spreading from the substantia nigra to other regions of the brain [11,12].

Several mechanisms are thought to play a role in the pathogenesis of PD, such as mitochondrial dysfunction, oxidative stress, neuroinflammation, protein aggregation, defective autophagy and environmental toxin-induced effects [13,14,15,16,17,18], with many of these mechanisms interlinked. For example, protein homeostasis can be disrupted by both increased protein aggregation and reduced protein clearance, while the generation of reactive oxygen species can be exacerbated by concurrent elevation of oxidative stress and impairment of mitophagy [15,18,19].

Rare mutations in a handful of genes are known to directly cause PD (Table 1), including α-synuclein (*SNCA*), leucine-rich repeat kinase 2 (*LRRK2*), PTEN-induced kinase 1 (*PINK1*), parkin (*PRKN*), parkinsonism-associated deglycase (*DJ-1*) and ATPase cation transporting 13A2 (*ATP13A2*) [20]. These genes were initially identified as monogenic causes of the disease primarily through linkage analysis of large families affected by PD. In addition, certain genetic risk factors are associated with the disease, including particular mutations in the glucocerebrosidase 1 (*GBA1*) and *LRRK2* genes [21,22,23]. For example, *GBA1* mutations are estimated to result in PD development in up to 30% of heterozygous carriers by the age of 80 [24,25,26,27].

Identification of additional genetic loci associated with PD has come from genome-wide association studies (GWASs), which test for allele frequency differences of hundreds of thousands of genetic variants across large numbers of individuals. A recent large multi-ancestry meta-analysis of PD GWASs identified 78 independent genome-wide significant loci (including in *SNCA* and *LRRK2*), 12 of which were not previously reported [28]. A PD GWAS meta-analysis was also carried out in 2019 [22], with the risk variants explaining 16–36% of PD heritable risk, highlighting that the etiology of the disease remains considerably unknown. It is likely that yet-to-be-identified PD genetic risk may be facilitated by larger-scale GWASs. However, given that the majority of GWAS variants lie in non-coding, regulatory regions of the genome [29] and PD concordance amongst monozygotic twins is only 17% [30], additional factors such as epigenetic variation are likely to play a pivotal role.

Epigenetics is the study of molecular processes that influence the regulation of gene expression without altering the underlying DNA sequence [31]. Key epigenetic mechanisms include DNA methylation, DNA hydroxymethylation, histone modifications, and actions mediated by non-coding RNAs such as microRNAs (miRNAs) [32]. Epigenetic processes often affect the accessibility of DNA to the transcription machinery, thus regulating gene expression levels [32]. Classic examples of epigenetic regulation include cell fate determination and specialization in embryonic development, X-chromosome inactivation and genomic imprinting [33]. Another example relevant to PD is neuronal function, where epigenetics is implicated in the synaptic plasticity of learning and memory formation [34,35]. Epigenetic dysregulation can lead to disease and has been particularly well studied in cancer [36]. In recent years, epigenetics has also been implicated in other neurodegenerative disorders, such as Alzheimer’s disease [37], and there is increasing evidence that epigenetics may play an important role in Parkinson’s.

Epigenetic changes might be influenced by genetic variation, and a few studies have started to integrate genetic and epigenetic information in PD [22,38,39,40,41]. On the other hand, epigenetic changes could be influenced by the environment. Mounting evidence suggests that long-lasting phenotypic changes can result from alterations in epigenetic make-up that are caused by environmental cues [33,42,43]. One such cue is stress, and several studies have demonstrated that stress, via epigenetic alterations, may impact the development, age of onset and progression of neurodegenerative and neuropsychiatric disorders, including PD, Huntington’s disease and depression [44,45,46]. In addition to stress, other factors such as prior head injuries and environmental contaminants (including toxins, heavy metals, pesticides and herbicides) have been associated with PD [47], and it is plausible that these factors also act via epigenetics to confer PD susceptibility. Finally, further evidence suggesting a link between the environment and epigenetic make-up comes from studies of monozygotic twins, which show that older twins have more epigenetic differences (and differential gene expression) compared to younger twins [48,49,50]. It is thought that these differences arise from twins spending an increasing amount of time in different environments over their lives. Therefore, the low concordance rate between twins for PD might be attributed to different environments, which, in turn, may cause distinct epigenetic changes that differentially influence disease susceptibility.

Importantly, epigenetic studies in Parkinson’s brains may not only unravel new mechanisms underlying disease development and progression but also uncover new drug targets and so advance novel therapeutic approaches. The development of new PD treatments would be particularly impactful as there are currently no treatments that can cure or modify the disease. The few treatments that do exist, including levodopa (l-DOPA), dopamine agonists and monoamine oxidase-B inhibitors, are symptomatic and only alleviate symptoms temporarily [51]. Moreover, these treatments become substantially less effective after only a few years and typically lead to problematic side-effects such as dyskinesia [52]. Therefore, new disease-modifying therapies that can slow, prevent or even reverse PD progression are urgently required.

In this review, we first summarize key findings related to epigenetics within the brain in Parkinson’s, covering DNA methylation, DNA hydroxymethylation, histone modifications and non-coding RNAs, focusing on the utility of epigenetics in identifying novel disease mechanisms. We then discuss the limitations of current epigenetic research in PD, the advantages of simultaneously studying genetics and epigenetics, the potential for epigenetic-based therapeutics and possible future directions.

## 2. DNA Methylation

DNA methylation is the most well studied epigenetic modification and consists of the addition of a methyl group to the 5-carbon position of a cytosine in a CpG dinucleotide (cytosine followed by guanine in the 5′-to-3′ direction) (Figure 2) [53,54]. This process is catalyzed by DNA methyltransferases (DNMTs), which are involved in either de novo (DNMT3a and 3b) or maintenance (DNMT1) methylation [54].

CpG sites (CpGs) are usually concentrated in CpG islands, which are commonly located in gene promoters [54]. Methylation of CpG islands within promoters has typically been associated with transcriptional silencing, either through blocking transcription factor binding or recruiting methyl-CpG binding proteins that themselves mediate transcriptional repression [53,55,56,57]. However, this is not always the case; for example, gene body DNA methylation has been associated with an increased level of gene expression and the regulation of alternative splicing [58,59,60].

### 2.1. Single-Gene Analyses

In Parkinson’s, initial DNA methylation studies mainly focused on genes already hypothesized to be associated with the disease (Table 2). Following bisulfite conversion of DNA, these studies mostly used either Sanger sequencing or pyrosequencing to measure DNA methylation. Pyrosequencing is a method of DNA sequencing that can be used to measure the degree of DNA methylation at individual CpG sites situated in close proximity [61]. The most extensively studied candidate gene in PD is *SNCA*, which encodes the α-synuclein protein. Point mutations in *SNCA* (including the Ala53Thr missense mutation) were the first identified genetic causes of PD [62,63,64], and *SNCA* locus amplifications (such as duplications and triplications) are also known to cause familial PD [65,66].

It is well established that expression can be regulated by DNA methylation levels within regulatory regions [53], and a number of studies have investigated whether DNA methylation within intron 1 of the *SNCA* gene is altered in PD. The majority of studies reported *SNCA* intron 1 DNA hypomethylation in PD brains [67,68,69,70]; however, others found no significant differences [38,71].

Jowaed et al. went further and examined the effect of inhibiting DNA methylation in SK-N-SH cells (a neuroblastoma cell line), which resulted in an increase in *SNCA* mRNA expression [67]. Thus, a plausible mechanism for the high levels of α-synuclein found in PD brains is DNA hypomethylation of the promoter region of *SNCA*.

A recent study highlighted that *SNCA* expression may be regulated by the binding of methyl-CpG binding protein 2 (MeCP2) to methylated CpG sites in *SNCA* intron 1 [70]. Using chromatin immunoprecipitation and qPCR in SK-N-SH cells, they showed that MeCP2 binds to *SNCA* intron 1 and that its binding is dependent on the degree of DNA methylation in intron 1. In the human cortex, *SNCA* intron 1 was hypomethylated in PD compared to controls, and the binding of MeCP2 also appeared to be lower in PD. In addition, CRISPR/Cas9 editing of SK-N-SH cells suggested that increased or decreased binding of MeCP2 to *SNCA* intron 1 is associated with a reduction or gain in α-synuclein expression, respectively [70].

Furthermore, based on a suggestion that α-synuclein may itself interact with the epigenetic machinery, Desplats et al. conducted a series of immunoprecipitation experiments using human brain homogenates that showed that α-synuclein does indeed interact with DNMT1 [69]. Next, they showed increased DNMT1 staining in the cytoplasm (compared to the nucleus) in the cells of PD brains compared to controls. Supported by further experiments in rat cells, they then concluded that DNMT1 may be sequestered to the cytoplasm by α-synuclein, which causes global DNA hypomethylation in PD brains [69].

A more recent study looked at *SNCA* intron 1 methylation separately in bulk frontal cortex, neuronal nuclei and glial nuclei (isolated using fluorescence-activated nuclear sorting) [72]. They observed borderline significant hypomethylation in neuronal nuclei but no significant differences in glial nuclei or bulk cortex. This may explain the mixed results of earlier studies where bulk samples were analyzed and highlights a potential advantage of profiling DNA methylation separately in individual cell types.

Another recent study started to take into account genetic variation by separating PD individuals with *GBA1* mutations from idiopathic PD individuals [73]. Significant hypomethylation of several CpGs in *SNCA* intron 1 in the frontal cortex was observed in PD-*GBA1* but not in idiopathic PD. This demonstrates that segregating PD cases according to genetic subtype might help uncover DNA methylation changes.

In addition to the *SNCA* gene, DNA methylation of other PD-related genes has also been investigated. These include *PRKN*, *PINK1* (both involved in mitochondrial function and mitophagy), *DJ-1* (involved in oxidative stress protection) and *MAPT* (which encodes the microtubule-associated protein tau that forms neurofibrillary tangles most notably in Alzheimer’s disease) [74,75,76,77]. For example, *MAPT* DNA methylation changes in PD varied according to brain region (hypermethylation in the cerebellum and hypomethylation in the putamen). The same study also suggested that the higher incidence of PD seen in males can possibly be explained by the higher *MAPT* DNA methylation levels observed in females and indicated that increased *MAPT* methylation was associated with later disease onset [77].

Some studies have investigated DNA methylation of genes that interact with key PD genes. For example, Su et al. examined peroxisome proliferator-activated receptor gamma coactivator 1α (*PGC-1α*) [78]. They found increased *PGC-1α* promoter methylation in the substantia nigra of PD individuals, along with a decrease in *PGC-1α* gene and protein expression. Previously, *PGC-1α* was shown to interact with parkin to regulate mitochondrial biogenesis and protect dopaminergic neurons [79]. It has also been associated with increased sensitivity to neurodegeneration [80].

It is worth noting that there are some key limitations of candidate gene studies. In general, small sample sizes are used, limiting their statistical power to detect significant methylation changes. Secondly, by their nature, they are based on a priori hypotheses, limiting the identification of novel genes epigenetically dysregulated in PD that have not been previously associated with neurodegenerative diseases. Thirdly, they are only testing one gene at a time rather than looking within the context of the whole genome (where it is now standard practice to correct for multiple testing) and are therefore subject to less statistical rigor. Therefore, rather than focusing on single genes, the field has shifted toward genome-wide approaches, which are discussed in the next section.

**Table 2 ijms-25-06168-t002:** Summary of DNA methylation studies in single genes in PD brains.

Gene(s)	Samples	Methylation Sites Studied	Key Findings	Reference/Year
*SNCA*	Substantia nigra and cortex (6 PD, 6 controls), putamen (6 PD, 8 controls)	23 CpGs in intron 1	Hypomethylation in all brain areas	[67]/2010
*SNCA*	Anterior cingulate cortex (11 PD, 8 controls), putamen (6 PD, 4 controls), substantia nigra (3 PD, 3 controls)	13 CpGs in intron 1	Hypomethylation in substantia nigra only	[68]/2010
*SNCA*	Brain homogenates (4 PD, 4 controls)	Gross methylation of intron 1	Hypomethylation	[69]/2011
*SNCA*	Cerebral cortex (12 PD, 12 controls)	Gross methylation of intron 1	No overall differences but methylation associated with rs3756063 SNP	[38]/2015
*SNCA*	Substantia nigra (8 PD, 8 controls)	23 CpGs in intron 1	No differences	[71]/2017
*SNCA*	Bulk frontal cortex (20 PD, 20 controls), sorted neuronal and glial nuclei (12 PD, 12 controls)	23 CpGs in intron 1	Bulk tissue and glial nuclei: No differences across all CpGs Neuronal nuclei: Hypomethylation in PD across all CpGs (borderline significant)	[72]/2021
*SNCA*	Frontal cortex (9 PD-*GBA1*, 11 idiopathic PD, 6 controls), putamen (6 PD-*GBA1*, 9 idiopathic PD, 6 controls), substantia nigra (8 PD-*GBA1*, 13 idiopathic PD, 3 controls)	17 CpGs (8 sites in intron 1 located further from TSS, 6 CpGs in intron 1 located close to TSS and 3 CpGs within promotor)	Frontal cortex: Hypomethylation in 5 CpGs located further from TSS in intron 1 in PD-*GBA1* compared to controls Hypomethylation in 2 CpGs in intron 1 in PD-*GBA1* compared to idiopathic PD No significant differences in any CpGs located close to TSS in intron 1 or in the promoter Putamen: No significant differences in intron 1 or the promoter Substantia nigra: 1 CpG in the promoter hypomethylated in idiopathic PD compared to both controls and PD-*GBA1* No significant differences in intron 1	[73]/2023
*SNCA*	Cortex (2PD, 2 controls)	23 CpGs in intron 1	Hypomethylation	[70]/2024
*PAD2*	Cortex (white matter) (2 PD, 0 controls)	Gross methylation of portion of intron 1, exon 1, and distal extension	No differences	[81]/2007
*TNF-α*	Substantia nigra and cortex (7 PD, 8 controls), striatum (3 PD, 2 controls)	10 CpGs in promoter	No differences	[82]/2008
*MAPT*, *PSEN1*, *APP* and *UCHL1*	Frontal cortex (8 PD, 17 controls)	*MAPT*: 20 CpGs near exon 0.41 + 31 CpGs in intronic regions *PESN1*: 26 CpGs at TSS *APP*: 18 + 16 CpGs in promoter regions *UCHL1*: 37 CpGs around TSS	No differences	[76]/2009
*PRKN*	Substantia nigra, cerebellum, occipital cortex (5 PD, 2 controls)	Gross methylation of promoter	No differences	[74]/2013
*MAPT*	Cerebellum, putamen, anterior cingulate cortex (28 PD, 12 controls)	6 CpGs within promoter/intron 1	Hypermethylation in cerebellum, hypomethylation in putamen, no significant differences in anterior cingulate cortex	[77]/2014
*ADORA2A*	Putamen (25 PD, 26 controls)	108 CpGs in 5′ untranslated region	Hypomethylation at 2 CpGs	[83]/2014
*PGC-1* *α*	Substantia nigra (10 PD, 10 controls)	Gross methylation of promoter	Hypermethylation (of mainly non-CpG dinucleotides) and decreased expression	[78]/2015
mtDNA D-loop	Substantia nigra (10 PD, 10 controls)	CpGSs and non-CpGs in D-loop	Hypomethylation in nearly all CpGs and non-CpGs	[84]/2016
*SNCA*, *LRRK2*, *PRKN*, *PINK1* and *DJ-1*	Substantia nigra, occipital cortex, parietal cortex (5 PD, 5 controls)	*SNCA*: 6–8 CpGs in promoter, 5 CpGs in exon 1 *LRRK2:* 9 CpGs in promoter, 9 CpGs in exon 1 *PRKN:* 4 CpGs in promoter, 9 CpGs in promoter overlapping with intron 1 *PINK1*: 5 CpGs in promoter/exon 1 *DJ-1*: 6 CpGs in promoter, 8 CpGs in promoter overlapping with exon 1	No significant differences at whole CpG islands, although differential methylation observed at some specific CpGs in *SNCA*, *PRKN* and *PINK1*	[75]/2018
*CYP2E1*, *TP73*, *C21ORF56* and *CDH13*	Cortex (14 PD, 10 controls)	*CYP2E1*: 10 CpGs (including promoter region) *TP73*: 5 CpGs *C21ORF56*: 2 CpGs *CDH13*: 1 CpG	Hypomethylation of *CYP2E1*, *TP73* and *C21ORF56* No difference in *CDH13* methylation	[85]/2022

Abbreviations: CpG site (CpG), transcription start site (TSS), single nucleotide polymorphism (SNP).

### 2.2. Epigenome-Wide Association Studies

Several epigenome-wide association studies (EWASs) have identified DNA methylation changes at both previously implicated and novel genes in PD brains (Table 3). Most of these studies used methylation arrays, which measure DNA methylation changes at many CpG sites throughout the genome. The first PD brain EWAS was performed by Kaut et al. using the Illumina HumanMethylation27 BeadChip, which covers 27,500 CpGs [86]. A few genes were found to be differentially methylated in either the putamen or cortex, but only *CYP2E1* (cytochrome P450 2E1) was significantly altered (hypomethylated) in both. This gene, which interestingly had already been implicated in PD pathology, encodes a protein that provides protection against xenobiotic exposure. The study also showed that decreased methylation of *CYP2E1* resulted in increased *CYP2E1* mRNA expression levels in the cortex.

Several subsequent studies used the Illumina HumanMethylation450 BeadChip, a substantially larger array that covers over 450,000 methylation sites. For example, Masliah et al. identified many differentially methylated CpGs between healthy and PD individuals [87]. Some of the top changing loci in this study were associated with four previously identified PD risk genes, including *MAPT*. It is worth noting that similar DNA methylation patterns were observed in both the brain and blood, suggesting that blood could potentially act as a surrogate for studying brain DNA methylation changes. This could have profound implications for developing easy-to-perform lab-based diagnostic tests for PD. Another study using the 450 BeadChip looked at DNA methylation changes in PD in relation to environmental exposures, namely plantation work and exposure to organochlorines [88]. Exposure to such pesticides has been associated with PD risk [89]. Since epigenetics may mediate environmental risk factors, the authors profiled DNA methylation changes in the brain and identified a few differentially methylated loci in PD individuals who were exposed to plantation work for 10+ years compared to non-exposed PD individuals [88]. Two of the loci differentially methylated between the organochlorine exposure groups were annotated to the *DNAJC15* gene, which is involved in protein translocation into the mitochondria and the regulation of the Hsp70 class of chaperones [90].

Over the last few years, there has been an increasing focus on looking at epigenetics in different brain cell types, and there are now a few PD studies that have isolated neuronal nuclei from the brain for DNA methylation profiling. Kochmanski et al. used the Infinium MethylationEPIC BeadChip (which contains probes for over 850,000 methylation sites) to profile DNA methylation changes in neuronal nuclei isolated from the parietal cortex [91]. They focused on sex-specific DNA methylation changes, given that there are known differences in disease risk, progression and severity between men and women, and sex-specific effects have not been investigated in previous EWASs. They found only three differentially methylated CpGs in males compared to 87 differentially methylated CpGs in females. The most significantly differentially methylated CpG in males was in the *PARK7* locus (hypomethylated) encoding the DJ-1 protein, while the most significantly differentially methylated CpG in females was within the *ATXN1* gene (hypermethylated). *DJ-1* hypomethylation may result in increased DJ-1 protein levels; however, *DJ-1* mutations in PD individuals have been linked with an apparent loss of function of DJ-1, leading to the accumulation of reactive oxygen species and decreased protection against oxidative stress [92].

Not all studies of genome-wide DNA methylation used arrays. For example, Marshall et al. used bisulfite padlock probe sequencing (a targeted bisulfite deep-sequencing method) to investigate genome-wide enhancer DNA methylation in neurons isolated from the prefrontal cortex [93]. A total of 1799 differentially methylated cytosines in enhancers were observed, which were predicted to target 2885 genes, including various PD risk genes (such as *DJ-1* and *PRKN*) as well as *TET2* (which plays a key role in DNA hydroxymethylation—see next section). Furthermore, *TET2* was also upregulated at the mRNA level in both neurons and bulk cytosol. This group also used the same technique to look at enhancer and promoter DNA methylation in neuronal nuclei isolated from each brain hemisphere [94]. They found that hemispheric asymmetry in DNA methylation was greater in PD individuals compared to controls and that this may be associated with the lateralization of disease symptoms.

Additionally, Gordevicius et al. used bisulfite padlock probe sequencing to profile DNA methylation of autophagy-lysosome pathway genes in the olfactory bulb and neuronal nuclei isolated from the prefrontal cortex [95]. In the olfactory bulb, there were 1142 differentially methylated sites, including *SNCA*, which was hypomethylated. In prefrontal cortex neurons, there were 70 differentially methylated sites; however, when only including PD brains with Braak stages 3–4, there were 110 differentially methylated sites, which significantly overlapped with the changes in the whole PD cohort. The authors suggest that alterations to DNA methylation might, therefore, begin earlier on in the disease before the prefrontal cortex is affected by Lewy body pathology.

Interestingly, some EWASs have identified DNA methylation changes of genes that interact with proteins encoded by key PD genes. For example, Dashtipour et al. observed robust hypermethylation of the *SNCAIP* gene, which encodes synphilin-1, in the cortex of PD individuals [96]. Synphilin-1 has been shown to interact with α-synuclein and regulate its degradation [97]. In another study, Young et al. identified hypermethylation of the *ARFGAP1* gene (which encodes a GTPase activating protein) in the dorsal motor nucleus of the vagus of PD individuals [98]. ARFGAP1 has been shown, both in vitro and in vivo, to interact with and regulate the activity of LRRK2 [99,100]. In particular, downregulation of *ARFGAP1* was shown to ameliorate the toxicity caused by *LRRK2* mutations [99,100], perhaps suggesting that *ARFGAP1* hypermethylation may have a protective role in PD brains.

Several of the aforementioned studies used Gene Ontology analysis to examine the functions of genes with altered methylation. Some of the key processes affected were neuronal development and differentiation, synaptic transmission, neurotransmitter transport, dopaminergic synapse, metabolism, immunity, bile acid secretion, cell cycle and cell signaling (including the Wnt pathway), all of which have been implicated in PD pathogenesis [87,91,93,94,98,101].

Some of the studies above report quite large numbers of differentially methylated CpGs; however, it is worth noting that not all studies accounted for multiple-testing correction. While several studies did report false discovery rate (FDR) corrected significance values, it may be more suitable to use a more stringent significance threshold of *p* < 9 × 10^−8^, which has been proposed to control for the false-positive rate in EPIC array studies [102]. This threshold was used in the Kochmanski et al. study [91].

**Table 3 ijms-25-06168-t003:** Summary of epigenome-wide association studies in PD brains.

Samples	Method	Key Findings	Reference/Year
Frontal cortex, cerebellum (399 healthy individuals)	Illumina HumanMethylation27 BeadChip	Differential methylation of 1 or more CpGs correlated with SNPs at PARK16/1q32, GPNMB/7p15, and STX1B/16p11 loci	[103]/2011
Cortex, putamen (6 PD, 6 controls)	Illumina HumanMethylation27 BeadChip	Cortex: Hypomethylation of *CYP2E1* and *PPP4R2* Putamen: Hypomethylation of *CYP2E1* and *LOC84245* Hypermethylation of *DEFA1* and *CHFR*	[86]/2012
Frontal cortex (5 PD, 6 controls)	Illumina HumanMethylation450 BeadChip	317 hypermethylated and 2591 hypomethylated CpGs	[87]/2013
Substantia nigra (39 PD, 13 controls)	Illumina HumanMethylation450 BeadChip	Hypermethylation of 1 CpG (cg10917602) associated with PD susceptibility	[104]/2016
Frontal cortex (12 PD, 12 controls)	Illumina HumanMethylation450 BeadChip	2794 differentially methylated CpGs in the frontal cortex of PD cases and 328 differentially methylated CpGs, majority hypomethylated. Clear pattern of *SNCAIP* hypermethylation	[96]/2017
Dorsal motor nucleus of the vagus, substantia nigra, cingulate gyrus (38 PD, 41 controls)	Illumina HumanMethylation450 BeadChip and Infinium MethylationEPIC BeadChip	234 differentially methylated regions in the dorsal motor nucleus of the vagus (including *ARFGAP1* hypermethylation), 44 in the substantia nigra and 141 in the cingulate gyrus	[98]/2019
Temporal lobe (13 PD with 0 years of plantation work, 4 PD with 10+ years of plantation work) (12 PD with 0–2 organochlorines, 4 PD with 4+ organochlorines)	Illumina HumanMethylation450 BeadChip	7 differentially methylated loci between PD individuals with 10+ vs. 0 years of plantation work exposure 8 differentially methylated loci between PD individuals with 4+ vs. 0–2 organochlorines in the brain 2 different loci annotated to *DNAJC15* which were differentially methylated (in both brain and blood) between the organochlorine exposure groups	[88]/2020
Prefrontal cortex neuronal nuclei (discovery cohort: 57 PD, 48 controls) (replication cohort: 26 PD, 31 controls)	Bisulfite padlock probe sequencing of enhancers and promoters (633,803 modified cytosines)	6207 CpGs in PD showing hemispheric asymmetry in DNA methylation (3894 CpGs showed a greater hemispheric asymmetry in PD compared to controls). These targeted 4691 genes, including PD risk genes. More DNA methylation and transcriptomic differences seen in hemisphere matched to symptom-dominant side Above findings validated in replication cohort 37 PD risk genes showing more hemispheric asymmetry in PD and/or greater differences in symptom-dominant hemisphere, including *SNCA*, *ITPKB*, *SATB1*, *ANK2* and *CAMK2D*	[94]/2020
Prefrontal cortex neuronal nuclei (57 PD, 48 controls) (22 PD Braak 3–4, 48 controls)	Bisulfite padlock probe sequencing (31,590 enhancers)	1799 differentially methylated cytosines in enhancers (mainly hypermethylated) 2172 differentially methylated cytosines in enhancers when comparing PD Braak stage 3–4 (prior to Lewy body pathology reaching the cortex) and controls Differentially methylated enhancers targeted 2885 genes, including 15 different PD risk genes and *TET2* 76 of the genes with dysregulated enhancers were also transcriptionally altered	[93]/2020
Olfactory bulb (9 PD, 14 controls), prefrontal cortex neuronal nuclei—discovery cohort (52 PD including 20 Braak stage 3–4, 42 controls), prefrontal cortex neuronal nuclei—replication cohort (13 PD, 15 controls)	Bisulfite padlock probe sequencing of autophagy-lysosome pathway genes (143,553 CpGs in olfactory bulb) (130,733 CpGs and 696,665 non-CpGs in prefrontal cortex discovery cohort) (110,397 CpGs in prefrontal cortex replication cohort)	Olfactory bulb: 1142 differentially methylated CpGs affecting 353 genes (mostly hypermethylated) (*SNCA* hypomethylation) Prefrontal cortex neuronal nuclei: 70 differentially methylated CpGs affecting 58 genes (mostly hypermethylated) in discovery cohort 110 differentially methylated CpGs affecting 87 genes in PD Braak stage 3–4 1131 differentially methylated CpGs affecting 341 genes in replication cohort	[95]/2021
Prefrontal cortex (27 PD, 26 controls)	Whole-genome bisulfite sequencing	No association between mitochondrial DNA methylation and disease status	[105]/2022
Cortex (14 PD, 10 controls)	Illumina HumanMethylation450 BeadChip	35 hypomethylated and 22 hypermethylated genes (not significant after *p*-value adjustment). Included 5 CpGs hypomethylated in *CYP2E1* and 6 CpGs hypomethylated in *C21ORF56*	[85]/2022
Sorted neuronal nuclei from parietal cortex (50 PD, 50 controls)	Infinium MethylationEPIC BeadChip	3 and 87 differentially methylated CpGs in males and females, respectively, including *PARK7* hypomethylation (males), *ATXN1* hypermethylation (females) and *SLC17A6* hypomethylation (females) 258 and 214 differentially methylated regions in males and females, respectively, including *NR4A2* (males) and *SLC17A6* (females) 1 differentially methylated region completely overlaps between sexes (annotated to *PTPRN2*)—hypermethylated in males and hypomethylated in females	[91]/2022
Primary motor cortex (40 PD, 38 controls)	Infinium MethylationEPIC BeadChip	3062 hypomethylated and 1251 hypermethylated CpGs 2.07 years of accelerated epigenetic age in PD compared to controls	[101]/2022
Prefrontal cortex neurons	Targeted bisulfite sequencing	667 differentially methylated genes, including 107 associated with stool butyrate levels	[106]/2022
Prefrontal cortex (19 PDD, 18 controls)	Infinium MethylationEPIC BeadChip	1151 differentially methylated CpGs (82% hypomethylated) 1 differentially methylated region in *OTX2* gene	[107]/2023

Abbreviations: epigenome-wide association study (EWAS), Parkinson’s disease dementia (PDD). Note: Of all the studies included in this table, only Kochmanski et al. [91] used the significance threshold of *p* < 9 × 10^−8^, which has been proposed to control for the false-positive rate in EPIC array studies.

## 3. DNA Hydroxymethylation

DNA hydroxymethylation is another, although less studied, DNA modification in which the ten-eleven translocation (TET) class of enzymes catalyze the oxidation of 5-methylcytosine to 5-hydroxymethylcytosine (by replacing the hydrogen atom at the carbon 5 position with a hydroxymethyl group) (Figure 3), which serves as an intermediate step in the DNA demethylation pathway [54,108,109]. DNA hydroxymethylation has recently gained interest as a potentially important epigenetic modification and is particularly abundant in neurons in the brain [110,111]. To date, only a handful of studies have looked at DNA hydroxymethylation in PD.

Using ELISA, Stöger et al. analyzed both DNA methylation and DNA hydroxymethylation in the cerebellum of 36 PD and 27 control individuals [112]. Although no significant differences were found in DNA methylation levels, PD individuals had significantly higher overall 5-hydroxymethylcytosine levels compared to the control group.

Kaut et al. examined DNA hydroxymethylation in the substantia nigra (eight PD and eight controls), cerebellum (eight PD and eight controls) and temporal gyrus (ten PD and ten controls) using an immunohistochemical approach [113]. They observed a significantly higher percentage of 5-hydroxymethylcytosine-immunoreactive cells in the PD cerebellum, supporting the Stöger study. However, no differences were seen in the substantia nigra or the neocortex. The lack of differences in these brain regions could be related to the small sample sizes used here.

Marshall et al. investigated whether *TET2* enhancer dysregulation affected hydroxymethylation levels in prefrontal cortex neurons [93]. Using hydroxymethylated DNA immunoprecipitation sequencing in 20 PD and 23 control individuals, they found that hydroxymethylation was considerably increased at gene bodies, promotors and enhancers in PD neurons and overlapped with the epigenetically modified enhancers. Experiments in mice showed that *TET2* silencing was neuroprotective. Therefore, TET2 could be a potential therapeutic target in PD [93].

Finally, Min et al. looked at genome-wide DNA hydroxymethylation in the substantia nigra of twelve PD and nine control individuals using hMe-Seal, which uses chemical labeling and affinity purification coupled with sequencing [114]. They identified 1800 hyperhydroxymethylated and 2319 hypohydroxymethylated regions (FDR < 0.05). In contrast, and perhaps surprisingly, when using methylated DNA immunoprecipitation sequencing to profile genome-wide DNA methylation, no differentially methylated regions were identified. Gene Ontology and Kyoto Encyclopedia of Genes and Genomes pathway analyses were also carried out, which highlighted that genes annotated to differentially hydroxymethylated regions were enriched for processes including neurogenesis and neuronal differentiation and were involved in multiple signaling pathways.

## 4. Histone Modifications

The third type of epigenetic mechanism that will be considered here is histone modifications. These are post-translational modifications to the N-terminal tails of histone octamers and include methylation, acetylation, phosphorylation, ubiquitination, SUMOylation and ADP-ribosylation [115,116]. These marks typically regulate chromatin structure and so alter the accessibility of DNA to the transcriptional machinery, with highly compacted chromatin (heterochromatin) associated with transcriptional repression and open chromatin (euchromatin) associated with transcriptional activation (Figure 4). Whether a given modification is repressive or active usually depends on the exact residue that is marked, although acetylation (which is the most well studied type of histone modification in PD) is normally associated with transcriptional activation [115,116].

So far, only a few studies have investigated histone modifications in PD brains (Table 4). For example, Gebremedhin and Rademacher used Western blotting to examine global H3 acetylation in the primary motor cortex and found a significant increase in H3K14 and H3K18 acetylation and a significant decrease in H3K9 acetylation, which correlated with both Lewy body stage and substantia nigra pigmentation scores [117]. In contrast, Harrison et al. found that H3K9 acetylation was increased in the substantia nigra of PD brains, with the size of the increase correlated with the Braak stage [118]. Further work is therefore needed to establish whether the direction of change in H3K9 acetylation is dependent on, for example, different brain regions.

A genome-wide study of histone acetylation was performed in the prefrontal cortex of PD brains, with a focus on H3K27 acetylation [119]. They identified 2877 H3K27-hyperacetylated regions and 14 H3K27-hypoacetylated regions, indicating a genome-wide dysregulation of H3K27 acetylation in the disease. Some of these regions were linked to known PD or neurodegenerative disease risk genes, including *SNCA*, *MAPT*, *PRKN*, *DJ-1* and amyloid precursor protein (*APP*). Perhaps interestingly, there was little correlation between H3K27 acetylation and gene transcription in the PD group (while a positive correlation was observed in the control group), suggesting that H3K27 acetylation may become decoupled from gene expression in PD.

Finally, Guhathakurta et al. studied a different histone mark—methylation—in *SNCA* regulatory regions and found that H3K4me3 was increased in PD bulk substantia nigra as well as neuronal nuclei isolated from this region [120]. Next, using CRISPR-Cas9 technology, they demonstrated that the reduction in H3K4me3 at the *SNCA* promoter leads to a decrease in α-synuclein levels both in SH-SY5Y cells and in idiopathic PD-iPSC-derived dopaminergic neurons.

**Table 4 ijms-25-06168-t004:** Summary of histone modification studies in PD brains.

Samples	Method	Key Findings	Reference/Year
Midbrain, cerebral cortex, cerebellar cortex (5 PD, 5 controls)	Western blotting and immunostaining	Midbrain: Increased acetylation of H2AK5, H2BK15, H3K9 and H4K5 (in 2–3 PD individuals) Downregulation of HDAC1, HDAC2, HDAC4, HDAC6 and SirT1 Higher proportion of acetylated midbrain dopaminergic neurons Cerebral cortex: Increased acetylation (H2AK5, H2BK15, H3K9 and H4K5) in 1 PD individual only Cerebellar cortex: Increased acetylation of H2BK15	[121]/2016
Primary motor cortex (9 PD, 8 controls)	Western botting	Increased acetylated H3–total H3 ratio Increased acetylated H3K14–total H3 ratio Increased acetylated H3K18–total H3 ratio Decreased acetylated H3K9–total H3 ratio	[117]/2016
Substantia nigra (8 early PD, 12 late PD, 10 controls)	Western blotting	Increased acetylation at H3K9 in late PD Correlation between level of histone acetylation and Braak stage	[118]/2018
Prefrontal cortex (global acetylation: 13 PD, 13 controls), Prefrontal cortex, striatum and cerebellar cortex (7 PD, 7 controls), Prefrontal cortex (discovery cohort: 17 PD, 11 controls), Prefrontal cortex (replication cohort: 10 PD, 11 controls)	Western blotting ChIP-seq (genome-wide H3K27 acetylation)	Prefrontal cortex: Increased global histone acetylation Increased acetylation at H3K27, H2BK15, H3K9/14, H3K56 and H4K12 No significant changes at H2AK5, H4K5 and H4K16 Striatum and cerebellar cortex: Increased acetylation at H3K27 Discovery study: 2877 H3K27-hyperacetylated regions (corresponding to 1434 genes) and 14 hypoacetylated regions (corresponding to 9 genes) Replication study: 2486 H3K27-hyperacetylated regions (corresponding to 946 genes) and 227 hypoacetylated regions (corresponding to 253 genes) 275 hyperacetylated genes (*DLG2* and *TNRC6B* most significant) and 2 hypoacetylated genes (*PTPRH*, *JUP*) replicated across both cohorts	[119]/2021
Substantia nigra (18 PD, 9 controls), substantia nigra neuronal nuclei (7 PD, 6 controls)	ChIP (H3K4me3, H3K27ac and H3K27me3)	Increased H3K4me3 at *SNCA* regulatory region No significant difference in H3K27ac Increased H3K4me3 at *SNCA* promoter/intron 1 Positive correlation between H3K4me3 and α-synuclein expression	[120]/2021
Substantia nigra (9 PD, 9 controls)	ChIP-seq (H3K27ac)	Identification of 2770 downregulated and 2910 upregulated cis-regulatory elements	[39]/2023

Abbreviations: chromatin immunoprecipitation (ChIP), histone deacetylase (HDAC), lysine (K).

## 5. Non-Coding RNAs

As with DNA methylation and histone modifications, non-coding RNAs are another epigenetic mechanism involved in the regulation of gene expression. These RNAs include miRNAs, long non-coding RNAs (lncRNAs), small nucleolar RNAs (snoRNAs), small inhibitory RNAs (siRNAs) and piwi-interacting RNAs (piRNAs) [122,123]. In PD, the most well studied of these are miRNAs, and this section will focus on miRNA and lncRNA expression studies.

miRNAs are small non-coding RNA molecules consisting of 20–22 nucleotides that bind to the 3′ untranslated region of target mRNAs, resulting in gene silencing by either translational repression or mRNA degradation (Figure 5) [124,125].

A number of studies have observed miRNA expression changes in the brains of PD individuals (Table 5). Several studies focused both on the identification of putative miRNAs dysregulated in PD and the subsequent functional analyses of those miRNAs in vitro. For example, suppression of miR-133b (the expression of which was significantly decreased in PD midbrain tissue) affected the expression of dopaminergic neuron markers in primary embryonic rat midbrain cultures. This suggested that miR-133b is involved in regulating the maturation and function of midbrain dopaminergic neurons [126]. In another example, overexpression of miR-126 (the expression of which was significantly increased in PD brains) impaired IGF-1/PI3K signaling in human SH-SY5Y cells, increased vulnerability to the neurotoxin 6-OHDA and reduced trophic support [127]. In yet another example, increased expression of a number of miRNAs (including miR-224, miR-379 and miR-26b in SH-SY5Y cells) resulted in a loss of expression of LAMP-2A and hsc70, key proteins involved in chaperone-mediated autophagy, as well as increased accumulation of α-synuclein, a substrate normally degraded by chaperone-mediated autophagy [128].

Multiple studies identified differentially expressed miRNAs that are directly or indirectly involved in the regulation of key PD genes, including miR-34b and miR-34c (depletion of which led to *DJ-1* and *PRKN* downregulation and *SNCA* upregulation), miR-205 (its downregulation resulted in *LRRK2* upregulation), miR-127-5p and miR-16-5p (both previously shown to regulate *GBA1* expression) [129,130,131,132,133]. The resulting altered expression of key PD genes mimics the upregulation or downregulation caused by genetic mutations in those genes, providing further evidence for their involvement in the pathogenesis and progression of the disease.

Interestingly, some miRNAs that showed altered expression in the brain were also found to be dysregulated in peripheral tissues. Some more commonly implicated miRNAs in both the brain and periphery include miR-30, miR-29, let-7, miR-485, miR-132 and miR-133b [126,132,134,135,136,137,138,139,140,141,142,143,144,145,146,147,148]. One study found 56 miRNAs dysregulated in both the brain and leukocytes [136]. Together, this highlights that peripheral samples may be suitable to use as a surrogate to identify brain miRNA changes, which can consequently help to identify novel pathways and mechanisms that are dysregulated in PD. Furthermore, the identification of PD-specific miRNA changes in easily accessible peripheral samples might lead to the development of new tests that would aid PD diagnosis.

Pathways involving some of the identified miRNAs include cell cycle regulation, apoptosis, axon guidance, neurogenesis, synaptic function, inflammation, oxidative stress, metabolism, ubiquitin/proteasome pathway, endocytosis and cell signaling (including insulin, NF-κB, p53 and mTOR) [127,134,136,138,149,150]. Interestingly, Ravanidis et al. found that certain processes (including ubiquitin-mediated proteolysis, circadian rhythm and axon guidance) are implicated in idiopathic but not genetic (*SNCA* or *GBA1*) PD cases [151].

Another important class of non-coding RNAs is lncRNAs, which are over 200 nucleotides long [152]. Altered lncRNA expression has been observed in PD brain regions, including the substantia nigra, cerebellum and cortex (Table 6). As with miRNAs, some studies provided functional insights into the roles that dysregulated lncRNAs might play in the pathogenesis of PD. For example, the expression of AL049437 was significantly upregulated in PD substantia nigra, and the reduction in its expression in SH-SY5Y cells led to increases in cell viability, tyrosine hydroxylase secretion and mitochondrial transmembrane potential and mass [153]. Another study found that the downregulation of six lncRNAs in PD substantia nigra and three lncRNAs in PD cerebellum was accompanied by a significant increase in *SNCA* mRNA levels in the substantia nigra and a significant decrease in *LRRK2* and *PINK1* mRNA levels in the substantia nigra and cerebellum. This highlights a putative correlation between the expression levels of lncRNAs and nearby PD genes [154].

**Table 5 ijms-25-06168-t005:** Summary of microRNA studies in PD brains.

Samples	Method	Key Findings	PD-Associated Pathophysiology	Reference/Year
Midbrain, cerebellum and cortex (3 PD, 5 controls)	RT-qPCR (panel of 224 miRNA precursors), RNase protection assay, qPCR and Northern blotting (including mature miR-133b)	Downregulation of miR-133b (precursor and mature)	Involved in the regulation of dopaminergic neuron maturation and function	[126]/2007
Amygdala (11 PD, 6 controls) Amygdala (13 PD, 12 controls), frontal cortex (14 PD, 21 controls), cerebellum (11 PD, 17 controls) and substantia nigra (7 PD, 6 controls)	miRCURY LNA™ *miRNA* microarray (17 miRNAs) RT-qPCR (miR-637, miR-34b, miR-34c)	Downregulation of miR-637 and miR-34c-5p in amygdala Downregulation of miR-34b and miR-34c validated in amygdala, frontal cortex, substantia nigra and cerebellum (only miR-34c significant in cerebellum) Could not confirm change in miR-637 expression (measured in amygdala)	Depletion of miR-34b and miR-34c led to *DJ-1* and *PRKN* downregulation and *SNCA* upregulation	[129]/2011
Substantia nigra and amygdala (6 PD, 5 controls)	RT-qPCR (8 miRNAs)	6 miRNAs upregulated in substantia nigra 2 miRNAs upregulated in amygdala	Target and associated with a reduction in *lamp-2a* and *hsc70* levels, key proteins involved in chaperone-mediated autophagy	[128]/2013
Frontal cortex (15 PD, 11 controls), striatum (5 PD and 4 controls)	RT-qPCR (miR-205)	Downregulation of miR-205	Downregulation of miR-205 resulted in *LRRK2* upregulation	[131]/2013
Substantia nigra (8 PD, 4 controls)	TaqMan low-density arrays (733 miRNAs) and TaqMan assays (miR-198, miR-548d, miR-385-5p and miR-135b)	10 miRNAs downregulated (including miR-135b) 1 miRNA upregulated (miR-548d)	Predicted target genes included *SNCA*, *PRKN*, *LRRK2*, *ATXN1*, *SNCAIP* and *GBA*	[137]/2014
Laser capture microdissected dopaminergic neurons (8 PD, 8 controls)	Human MicroRNA TaqMan Arrays (379 miRNAs)	Dysregulation of miRNA expression profile Upregulation of miR-126	miR-126 overexpression impaired IGF-1/PI3K/AKT signaling	[127]/2014
Midbrain tissue, laser capture microdissected dopaminergic neurons (5 PD, 8 controls)	RT-qPCR (miR-133b)	Downregulation of miR-133b in midbrain tissue No differences in miR-133b levels in dopaminergic neurons		[146]/2014
Putamen (25 PD, 26 controls)	RT-qPCR (miR-34b and c)	Downregulation of miR-34b, particularly in early stages	Adenosine A2A receptor (*A_2A_R*) identified as potential target of miR-34b	[83]/2014
Laser capture microdissected dopaminergic neurons (8 PD, 8 controls)	Human MicroRNA TaqMan Arrays	109 miRNAs upregulated (miR-132 significantly upregulated) 50 miRNAs downregulated 14 significantly differentially expressed miRNAs associated with target genes Trend toward upregulation in males and downregulation in females	Targets associated with several aspects of PD pathogenesis, including cellular function and dopaminergic neuron identity	[134]/2015
Prefrontal cortex (29 PD, 33 controls)	Small RNA sequencing—Illumina HiSeq 2000 (911 miRNAs)	125 differentially expressed miRNAs (downregulation of miR-10b-5p)	Including miR-127-5p and miR-16-5p, both previously shown to regulate *GBA1* expression	[132]/2016
Putamen (12 PD—mostly l-DOPA treated, 12 controls)	Human v2 miRNA expression assay kit (800 miRNAs) and RT-qPCR (4 miRNAs)	6 miRNAs upregulated 7 miRNAs downregulated Upregulation of miR-3195 and miR-204-5p Downregulation of miR-155-5p and miR-219-2-3p	miRNAs associated with inflammatory response and oxidative stress	[138]/2016
Amygdala (14 PD, 7 controls)	RNA-seq	42 differentially expressed miRNAs in premotor-stage PD compared to controls 103 differentially expressed miRNAs in motor-stage PD compared to controls		[155]/2016
Anterior cingulate gyrus (22 PD, 10 controls)	TaqMan miR array (744 miRNAs) RT-qPCR (13 miRNAs)	43 miRNAs upregulated 5 miRNAs upregulated	13 of these each predicted to regulate at least one of *DJ-1*, *PRKN*, *PINK1*, *LRRK2*, *SNCA* or *HTRA2* Predicted to each regulate at least one of *SNCA*, *PRKN* or *LRRK2* and additional genes involved in normal cellular function	[135]/2016
Prefrontal cortex (29 PD, 36 controls)	RNA-seq (99 novel miRNAs)	Upregulation of miR-46 and miR-236 Downregulation of miR-225		[156]/2016
Prefrontal cortex (29 PD, 33 controls)	RNA-seq	321 differentially expressed miRNAs		[136]/2017
Substantia nigra (6 PD, 5 controls)	RT-qPCR (miR-7)	Downregulation of miR-7	Depletion of miR-7 results in increased α-synuclein expression, dopaminergic neuron loss and reduced striatal dopamine content	[157]/2017
Cingulate gyrus (8 PD, 8 controls)	RNA-seq	44 miRNAs upregulated 55 miRNAs downregulated		[158]/2018
Substantia nigra (4 PD, 4 controls)	In situ hybridization (miR-425)	Downregulation of miR-425	miR-425 deficiency triggers necroptosis of dopaminergic neurons	[159]/2019
Prefrontal cortex (15 PD, 10 controls)	RT-qPCR (10 miRNAs)	3 miRNAs downregulated (miR-124, miR-144 and miR-218)	Target *KPNB1*/*A3*/*A4*, which were all upregulated in PD brains. Inhibition of these miRNAs activates NF-κB signaling	[149]/2020
Midbrain (19 PD, 12 controls)	Small and total RNA-seq, RT-qPCR (4 miRNAs)	4 miRNAs upregulated (miR-539-3p, miR-376a-5p, miR-218-5p, miR-369-3p)	Targets of miR-369-3p (*GTF2H3*) and miR-218-5p (*RAB6C*) downregulated	[160]/2022
Superior temporal gyrus (214 PD, 47 controls)	TaqMan Advanced miRNA Assays (10 miRNAs)	3 miRNAs downregulated (miR-132-3p, miR-132-5p and miR-129-5p)	miR-132-3p/-5p significantly associated with α-synuclein Braak stage and may interact with *SNCA* mRNA	[161]/2022
Midbrain (5 PD, 5 controls)	RT-qPCR (miR-132-3p)	Upregulation of miR-132-3p	*GLRX* identified as potential miR-132-3p target, *GLRX* mRNA and protein expression decreased in PD	[162]/2022
Middle frontal gyrus (16 PD Braak stage 4, 9 PD Braak stage 5–6, 19 PDD (Braak stage 5–6), 19 controls)	RNA-seq RT-qPCR (let-7e-3p, miR-424-3p and miR-543)	9 miRNAs downregulated 3 miRNAs upregulated (combined PD groups) Upregulation of let-7e-3p in PD with Braak 5–6 compared to PDD in both gray and white matter Upregulation of miR-424-3p in PD with Braak 5–6 in both gray and white matter compared to controls, and in PD in gray matter compared to PDD Upregulation of miR-543 in PD compared to controls in white matter only	*SIRT1* identified as potential miR-543 target	[150]/2022

Abbreviation: reverse transcription polymerase chain reaction (RT-qPCR).

**Table 6 ijms-25-06168-t006:** Summary of long non-coding RNA studies in PD brains.

Samples	Method	Key Findings	PD-Associated Pathophysiology	Reference/Year
Substantia nigra, amygdala (5 PD, 5 controls)	RT-qPCR (3 lncRNAs)	3 lncRNAs upregulated in amygdala (RP11-462G22.1, RP11-79P5.3 and U1) RP11-462G22.1 and RP11-79P5.3 upregulated in substantia nigra		[163]/2014
Anterior cingulate gyrus neurons (20 PD, 10 controls)	RT-qPCR (90 lncRNAs)	4 lncRNAs upregulated (lincRNA-p21, Malat1, SNHG1 and TncRNA) Downregulation of H19 lncRNA		[164]/2017
Substantia nigra (11 PD, 14 controls)	Affymetrix Human Genome U133A Array (698 lncRNAs)	42 lncRNAs upregulated (AL049437 most significantly upregulated) 45 lncRNAs downregulated (AK021630 most significantly downregulated)	Reduction in AL049437 expression led to increases in cell viability, tyrosine hydroxylase secretion and mitochondrial transmembrane potential and mass	[153]/2017
Substantia nigra and cerebellum (9 PD, 8 controls)	RT-qPCR (6 lncRNAs)	6 lncRNAs downregulated in substantia nigra 3 lncRNAs downregulated in cerebellum (AK127687, UCHL1-AS1, MAPT-AS1)	Accompanied by increased *SNCA* mRNA levels in the substantia nigra and decreased *LRRK2* and *PINK1* mRNA levels in both brain regions	[154]/2019
Substantia nigra (29 PD, 24 controls)	RT-qPCR (NEAT1 lncRNA)	Upregulation of NEAT1	Neuroprotective agents induce NEAT1 upregulation	[165]/2019
Superior frontal gyrus (23 Braak Lewy body stage 0 controls, 61 PD/PDD/incidental Lewy body disease—subdivided into 19 Braak Lewy body stage 1–4, 19 Braak Lewy body stage 5, 23 Braak Lewy body stage 6)	RNA-seq	Differential expression of 34 lncRNAs between groups		[166]/2023
Substantia nigra (57 PD, 43 controls)	Bioinformatics analysis of microarray data	37 lncRNAs upregulated 68 lncRNAs downregulated		[167]/2023

Abbreviation: long non-coding RNA (lncRNA).

## 6. Discussion

Figure 6 provides a summary of the potential roles of DNA methylation, histone modifications and miRNAs in PD from the studies discussed, including some of the key genes and biological processes affected.

### 6.1. Limitations of Current Studies

Firstly, several of the candidate gene DNA methylation and miRNA studies used small sample sizes and might, therefore, lack the power to identify significant epigenetic changes. This could explain some of the discrepancies between the findings reported. For example, while several studies observed *SNCA* intron 1 hypomethylation [67,68,69,70], not all studies have replicated this [38,71]. Even when the same brain regions were analyzed, for example, substantia nigra, cortex or putamen, there were discrepant findings between studies [38,67,68,71]. There are also some inconsistent findings between different miRNA studies. For example, some of the key dysregulated miRNAs identified by earlier studies include miR-34b/c, miR-133b and miR-205 [126,129,131]. However, several subsequent studies failed to observe a difference for some or all of these miRNAs [132,134,135,137,155]. Therefore, performing studies using greater sample sizes may help provide more conclusive answers where there are conflicting findings.

Furthermore, not all miRNA studies used the same techniques; the earlier studies (up to 2014) almost exclusively used RT-qPCR, whereas some of the later studies used microarrays and RNA-seq. Therefore, differences in findings between studies could be influenced by the use of different laboratory approaches as well as small sample sizes. A few of the miRNA brain studies used more than one technique to analyze the same miRNAs, for example, using arrays followed by RT-qPCR [129,135,138]. This type of approach should be encouraged to validate initial findings and help prioritize miRNAs for further investigation in future studies. Another factor to consider is the disease stage of the samples; for example, Miñones-Moyano et al. observed miR-34b/c downregulation in the early stage of the disease [129], while Cardo et al. failed to replicate this finding using advanced-stage brains [137]. Pantano et al. compared miRNA expression in both early and late-stage PD to controls and suggested that miRNA profiles vary according to disease stage [155]. Therefore, the variation in findings between studies could also be influenced by differences in the disease stages of the brains profiled.

Furthermore, it is well known that PD pathology differs between cell types. In particular, in addition to neurons (whose role in PD via α-synuclein accumulation and dopaminergic neuron loss is well established) [9], multiple studies have shown that oligodendrocytes and other types of glial cells are also involved [168,169,170,171,172,173,174,175,176]. However, despite this, almost all PD epigenetic studies performed to date have only analyzed bulk brain tissue. This is a substantial drawback since results then reflect changes in a mixture of both neuronal and different non-neuronal cell types, and it is known that epigenetic profiles are highly cell type-specific [177,178]. Furthermore, results may be confounded by the proportions of the different cell populations present, which may vary from sample to sample. Therefore, bulk brain tissue studies mask the epigenetic disease signatures of distinct cell types, hampering both mechanistic understanding and the search for novel drug targets.

Novel protocols for fluorescence-activated nuclear sorting are now emerging, allowing the study of epigenetics in individual cell types [179,180,181]. In the last few years, there have been initial studies in PD that isolated neuronal nuclei from human brain tissue and detected genome-wide DNA methylation changes in these nuclei [93,94,95,106]. Since Gu et al. observed *SNCA* hypomethylation in neuronal nuclei but not in glial nuclei or bulk frontal cortex, perhaps some of the discrepancies in the earlier candidate gene studies could be attributed to differences in cell proportions [72]. Similarly, one study compared miRNA expression changes in isolated neurons to bulk brain tissue. While miR-133b was downregulated in PD midbrain tissue, no change in the level of this miRNA was observed in laser capture microdissected dopaminergic neurons [146], perhaps indicating that the difference found in bulk brain tissue is driven by glial cells. Separating cell populations in epigenetic studies could also make the signals less noisy due to the enrichment of a certain cell type, so studies will likely require smaller sample sizes to achieve the same statistical power compared to bulk brain studies. Cell type-specific approaches are crucial going forward to provide a more detailed picture of the role of epigenetics in PD.

In addition to this, studies so far have generally not considered different genetic subtypes of PD. Smith et al. reported *SNCA* intron 1 hypomethylation in PD individuals with *GBA1* mutations but *SNCA* promotor hypomethylation in idiopathic PD [73], highlighting the existence of potentially distinct epigenetic profiles between PD individuals with *GBA1* mutations and idiopathic PD cases. These findings could, perhaps, explain why previous studies (which do not separate PD-*GBA1* and idiopathic PD) have led to mixed results regarding *SNCA* methylation. This demonstrates the putative importance of separating the two groups in future studies as well as further stratifying by other known PD genes to gain a better understanding of DNA methylation changes in the disease.

### 6.2. Utility of Combining Genetics and Epigenetics Studies

Integrating genetic and epigenetic information is essential to further understand the etiology of and mechanisms involved in diseases. The majority of GWAS variants are found in regulatory regions, and it is known that epigenetic changes can be genetically influenced [182]. Genetic variants that are associated with the levels of DNA methylation at specific positions in the genome are called methylation quantitative trait loci (mQTLs).

One study examined the effects of PD risk variants in the *SNCA* locus on intron 1 methylation levels and identified rs3756063 as an mQTL associated with *SNCA* intron 1 methylation [38]. This SNP was found to be localized 30 bp upstream of a GATA transcription factor binding motif within *SNCA* intron 1 and additionally alters a CpG dinucleotide itself, suggesting it may affect the interaction between GATA and the *SNCA* gene. GATA transcription factors have previously been shown to regulate *SNCA* expression [183].

Other studies have examined the overlap between SNPs in the *SNCA* gene and epigenetic marks. Sharma et al. found that several SNPs distributed throughout the *SNCA* gene (which were previously associated with PD) were located within or near peaks of certain histone modification marks and DNA binding motif sequences [40]. In another example, certain SNPs associated with PD were present in enhancer (H3K27-acetylated) regions in the *SNCA* and *PARK16* loci [41].

In addition, summary-data-based Mendelian randomization (SMR) is an approach that combines summary-level GWAS data with DNA methylation (mQTL) or gene expression (eQTL) data to identify genes with methylation or expression levels that are either pleiotropically or causally associated with a particular trait [184]. SMR works by triangulating SNP association statistics with the phenotypic trait on SNP association statistics with methylation/expression to provide an estimate of the effect of a genetically determined increase in DNA methylation or gene expression on the disease phenotype [184,185]. The 2019 PD GWAS meta-analysis by Nalls et al. used SMR and found a significant association between the methylation or expression of 151 genes (around PD risk variants) and a potential causal change in PD risk [22].

Finally, multiomic approaches can be used to aid in the identification of genes involved in PD and provide more insights into the roles of specific cell types. A recent study combining chromatin accessibility (snATAC-seq), transcriptomic (snRNA-seq), histone modification (bulk H3K27ac ChIP-seq) and GWAS data investigated cell type-specific disruptions in the PD substantia nigra [39]. A total of 2770 downregulated and 2910 upregulated cis-regulatory elements (cREs) were identified in this brain region in PD. Integration of the dysregulated cREs and PD GWAS SNPs led to the identification of 656 target genes (potential PD candidate genes) in specific cell types (including dopaminergic neurons, GABAergic neurons, oligodendrocytes, oligodendrocyte precursor cells, astrocytes, microglia, endothelial cells and pericytes). Over 50% of these target genes were assigned to only one or two cell types, indicating a high degree of cell type specificity. Oligodendrocyte and microglial cREs were strongly enriched in more than one PD GWAS, and the authors suggest that these are the two key cell types associated with the disease (also supported by RNA-seq data). Active regulatory associations involving known PD genes were annotated to different cell types. For example, certain PD GWAS SNPs were specific to dopaminergic neurons, *SNCA* was involved in most cell types, and *MAPT* was mainly associated with oligodendrocytes, OPCs and astrocytes. In addition, it was found that transcription factor binding is disrupted by PD GWAS variants in cis-regulatory regions. The authors conclude that common risk variants exhibit cell type-specific regulatory functions and that the combination of both genetic predisposition and epigenetic alterations contributes to PD-related cellular processes and pathogenesis.

This study highlights the potential advantages of integrating genetic, epigenetic and transcriptomic information to understand more about the roles of PD risk variants in particular cell types. Future multiomic studies should be performed, perhaps in different brain regions, in addition to incorporating DNA (hydroxy)methylation and proteomic data to build a more comprehensive understanding of PD risk and mechanisms, including cell type-specific changes.

### 6.3. Epigenetic Therapies

An understanding of epigenetics in Parkinson’s could lead to new drug targets and potential novel therapeutic approaches. Such therapies could take advantage of the fact that epigenetic changes are reversible, and thus, it might be plausible to correct the altered epigenetic modifications in PD.

Research in cancer shows that therapies based on correcting DNA methylation or histone modification changes are possible. Here, several drugs are in clinical trials, and a few are FDA-approved. This includes DNMT inhibitors such as azacitidine, approved for the treatment of myelodysplastic syndrome and acute myeloid leukemia, as well as HDAC inhibitors such as panobinostat, a third-line treatment for multiple myeloma [186].

A number of studies in cell and animal models of PD have found that HDAC inhibitors such as sodium butyrate and valproic acid (both increasing acetylation) can be neuroprotective, reduce α-synuclein levels, increase autophagy, restore striatal dopamine levels and attenuate motor and non-motor changes [187,188,189,190,191,192,193,194,195,196,197,198]. However, at present, it is unclear whether further increasing histone acetylation would be beneficial in individuals with PD, as the studies to date have predominantly reported the presence of widespread hyperacetylation in PD brains.

An alternative and more specific approach may be to module the expression of genes that are epigenetically dysregulated in PD. For example, short hairpin RNA mini-circles targeting α-synuclein were delivered using exosomes into a PD mouse model, which reduced α-synuclein aggregation and dopaminergic neuron loss and improved disease symptoms [199]. CRISPR/Cas9-based epigenome editing technologies can also be used here. A recent study trialed a lentiviral vector-based therapy containing a deactivated form of CRISPR/Cas9 which targeted *SNCA* intron 1 specifically in dopaminergic and cholinergic neurons [200]. Introducing the therapy into iPSC-derived dopaminergic and cholinergic neurons from a *SNCA* triplication patient led to the downregulation of *SNCA* mRNA and protein in these specific cell types as well as increasing neuronal viability and mitochondrial function. In another example, the observed reduction in α-synuclein levels following the demethylation of H3K4 at the *SNCA* promoter using CRISPR/Cas9 [120] suggests that a possible future PD treatment avenue might involve region-specific methylation or demethylation of DNA or certain histones to reverse epigenetic changes in the disease.

Furthermore, the expression of miRNAs can be modulated by either miRNA mimics or antagomirs. For example, Hu et al. observed a decrease in miR-425 expression in the human substantia nigra, while injection of miR-425 mimics into the substantia nigra of MPTP-treated mice attenuated necroptosis activation, protected against dopaminergic neuron loss, increased striatal dopamine levels and reduced deficits in motor activity [159]. In another example, Zhou et al. found that miR-103a-3p was predicted to bind parkin and was upregulated in both MPP+-induced SH-SY5Y cells and MPTP-induced mouse models [201]. This miRNA was shown in another study to be upregulated in the blood of l-DOPA-treated PD patients [144]. Zhou et al. observed that transfection of a miR-103a-3p antagomir into both the cell model and striatum of the mouse model led to the upregulation of parkin and improved mitophagy [201]. Additionally, it resulted in neuroprotection in the mouse model, preventing the loss of TH-positive cells in the substantia nigra. This suggests that modulating the levels of these miRNAs described may be beneficial in PD. However, finding an appropriate miRNA to target may be difficult as a single miRNA can have multiple mRNA targets; therefore, downstream and off-target effects need to be considered [202].

To sum up, epigenetic-based therapies are an exciting prospect for PD and other neurodegenerative disorders. However, this is a rather long way off, with further advances in mechanistic understanding required and some challenges needing to be overcome.

### 6.4. Future Directions

There are a number of avenues that could be explored in the near future to further our understanding of PD epigenetics. Firstly, larger-scale EWASs should be performed to both verify previously found and identify novel epigenetic changes in PD, in particular isolating different brain cell types and analyzing cell type-specific epigenetic changes. Secondly, future work should also investigate the interplay between genetics and epigenetics in PD. In particular, multiomic approaches might help shed light on the pathogenesis, for example, by identifying biologically relevant genes and cell types as well as potential therapeutic targets. Additionally, the concurrent screening of brain samples for genetic and epigenetic alterations could help determine whether the presence of epigenetic modifications combined with genetic mutations increases disease risk. Furthermore, future work should involve stratifying individuals by genetics to compare epigenetic profiles between different genetic subtypes of the disease. Thirdly, human postmortem brain studies are the most direct method for studying epigenetic modifications in neurodegenerative diseases. However, in postmortem tissue, it is difficult to establish whether epigenetic changes play a causal role in the disease. Studies comparing epigenetic alterations in PD samples of different Braak stages, including very early-stage individuals, may help address this question. In addition, this could allow the study of epigenetic dysregulation over the course of disease progression. Finally, continued experiments in animal models and iPSC-derived cells are also crucial, the latter of which more closely resemble affected individuals. Therefore, functional studies in cell and animal models should help investigate (1) whether epigenetic dysregulation is a cause or consequence of PD pathology and (2) whether epigenetic alterations can be targeted for novel therapeutic approaches. Furthermore, CRISPR/Cas9-based epigenome editing technologies could be utilized to introduce or remove a particular epigenetic modification at specific sites in the genome in order to evaluate its exact biological consequences.

## Figures and Tables

**Figure 1 ijms-25-06168-f001:**
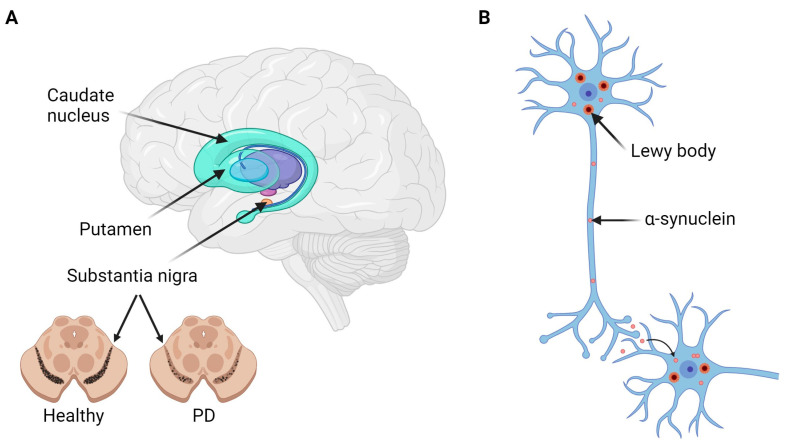
Overview of PD pathology. Abbreviations: Parkinson’s disease (PD). (**A**) Schematic of the brain highlighting the substantia nigra, putamen and caudate nucleus (the latter two are both part of the striatum). Dopaminergic neurons are depleted in the PD substantia nigra, as shown below. (**B**) Schematic of two neurons affected by α-synuclein and Lewy body pathology, which have spread from one neuron to the next. Created with BioRender.com.

**Figure 2 ijms-25-06168-f002:**
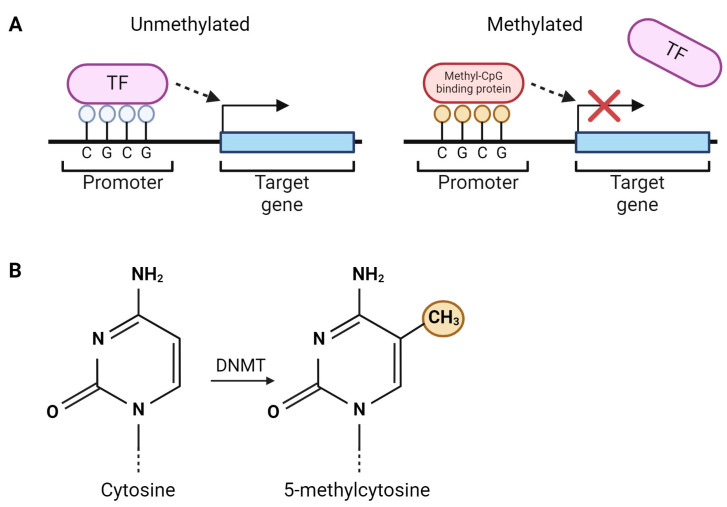
DNA methylation overview. Abbreviations: transcription factor (TF), DNA methyltransferase (DNMT). (**A**) Schematic of an unmethylated (left) and methylated (right) CpG island in a gene promotor and target gene. When unmethylated, transcription factors can typically bind, allowing the gene to be expressed. When methylated, transcription factor binding is typically prevented, and a methyl-CpG binding protein is recruited. Both of these events can lead to the gene being silenced. (**B**) Structural formulas of cytosine and 5-methylcytosine. DNA methylation is catalyzed by DNMT enzymes. Created with BioRender.com.

**Figure 3 ijms-25-06168-f003:**
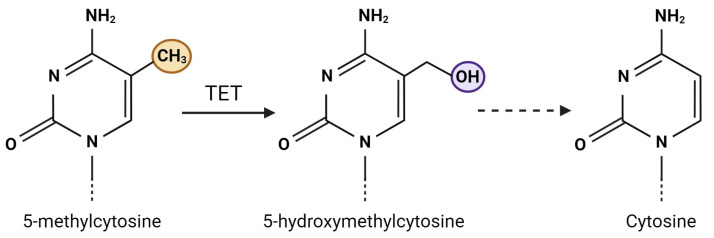
DNA hydroxymethylation overview. Abbreviations: ten-eleven translocation (TET). Structural formulas depicting the conversion of 5-methylcytosine to 5-hydroxymethylcytosine, catalyzed by TET. 5-hydroxymethylcytosine can also serve as an intermediate in the DNA demethylation pathway and be converted back to cytosine. Created with BioRender.com.

**Figure 4 ijms-25-06168-f004:**
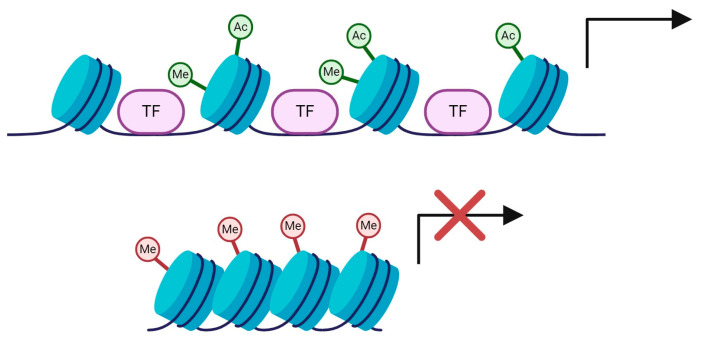
Histone modifications overview. Abbreviations: acetylation (Ac), methylation (Me). The top picture depicts an open chromatin region where the histone octamers contain active acetylation and active methylation marks, resulting in transcription factor binding and gene expression. The bottom picture depicts a compacted chromatin region where the histone octamers contain repressive methylation marks. This prevents transcription factors from binding, resulting in gene silencing. Created with BioRender.com.

**Figure 5 ijms-25-06168-f005:**
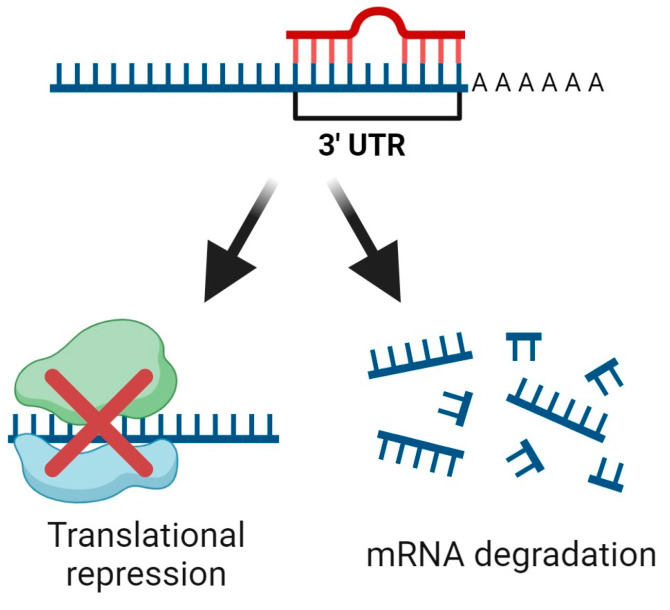
microRNAs overview. Abbreviations: microRNA (miRNA), untranslated region (UTR). Schematic illustrating how a miRNA binds to the 3′UTR of a target mRNA, which results in gene silencing either by translational repression or mRNA degradation. Created with BioRender.com.

**Figure 6 ijms-25-06168-f006:**
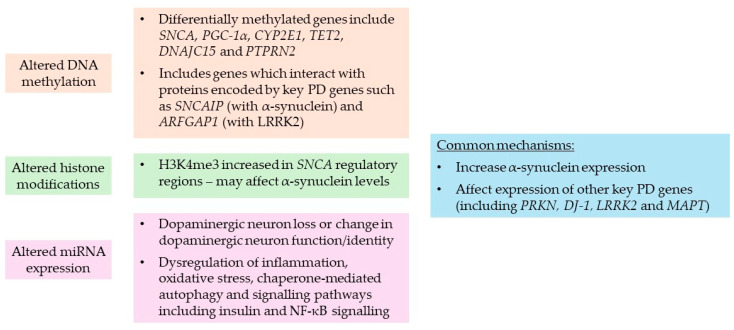
Summary of how epigenetics may affect PD pathogenesis. This figure highlights the potential roles of DNA methylation, histone modifications and miRNA expression in PD. Studies have related all three epigenetic mechanisms to an increase in α-synuclein expression and changes in the expression of multiple PD risk genes.

**Table 1 ijms-25-06168-t001:** Key rare mutations that cause PD.

Gene Name	Abbreviation	Mode of Inheritance
α-synuclein	*SNCA*	Autosomal dominant
leucine-rich repeat kinase 2	*LRRK2*	Autosomal dominant
PTEN-induced kinase 1	*PINK1*	Autosomal recessive
parkin	*PRKN*	Autosomal recessive
Parkinsonism-associated deglycase	*DJ-1*	Autosomal recessive
ATPase cation transporting 13A2	*ATP13A2*	Autosomal recessive

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
