# Peer review of "An Overview of Epigenetic Changes in the Parkinson’s Disease Brain"

_ijms, 2024, doi:10.3390/ijms25116168_

Round 1

Reviewer 1 Report

Comments and Suggestions for Authors

Overall, a well-written and very organized review that will be useful for all readers in the field. Please find attached minor changes/suggestions for consideration. 

I thank the journal for giving me an opportunity to review the article titled “An overview of epigenetic changes in Parkinson’s disease brains” by Klokkaris and Migdalska-Richards. Overall, a well-written and organized review that will be useful for all readers in the field. Some minor changes/suggestions below for consideration –

1.      Lines 37-46: a simplified figure showing the details of the paragraph (anatomical locations etc) would be useful for the readers.

2.      A table listing the rare mutations (line 57-74) would be helpful for a quick visual review.

3.      Lines 75-87 would be easier if supported with a simplified cartoon/figure (for example something like in PMID: 29953914, Figure 3).

4.      Simple figures/cartoons for each of the epigenetic path mentioned in this paper (sections 2, 3, 4 and 5 for example) would be useful for non-specialist readers.

5.      For all tables, please include the year as well in the last column “Reference/Year”

6.      Table 3 (and all other tables), please include abbreviations in the end (also mention what “K” stands for (lysine residue) in the table.

7.      Please include a statement in section 5 (Non-coding RNAs) that the review focuses mainly on data collected on miRNA and lncRNA.

8.      Very minor language edits required – for example, consecutive paragraphs (lines 539 and 549) start with “in addition” (which does not read well). Suggested title change for consideration - “An overview of epigenetic changes in the Parkinson disease brain”

9.      Section 6.3, all points needn’t start with a separate line/para, please condense it all and make it continuous.

All in all, a detailed and very well-organized review paper.

Comments on the Quality of English Language

Very minor edits needed. 

Author Response

We thank the Reviewer for taking their time to revise our manuscript and for their useful comments. Please find our answers to the Reviewer's comments below.

REVIEWER 1:
Comments and Suggestions for Authors

Overall, a well-written and very organized review that will be useful for all readers in the field. Please find attached minor changes/suggestions for consideration. 

I thank the journal for giving me an opportunity to review the article titled “An overview of epigenetic changes in Parkinson’s disease brains” by Klokkaris and Migdalska-Richards. Overall, a well-written and organized review that will be useful for all readers in the field. Some minor changes/suggestions below for consideration –

1.      Lines 37-46: a simplified figure showing the details of the paragraph (anatomical locations etc) would be useful for the readers.

A simplified figure (figure 1) was created to show brain regions involved and give visual representation of synuclein pathology.

2.      A table listing the rare mutations (line 57-74) would be helpful for a quick visual review.

A table (table 1) listing the key rare mutations was created.

  1. Lines 75-87 would be easier if supported with a simplified cartoon/figure (for example something like in PMID: 29953914, Figure 3).

Four separate figures (Figures 2-5), each presenting an overview of epigenetic mechanism discussed in relevant subsections, were created.

  1. Simple figures/cartoons for each of the epigenetic path mentioned in this paper (sections 2, 3, 4 and 5 for example) would be useful for non-specialist readers.

Figures 2-5 were created to give explanation of DNA methylation, DNA hydroxymethylation, histone modifications and miRNAs respectively.

  1. For all tables, please include the year as well in the last column “Reference/Year”

The year in reference column was added for all tables (in general studies are in oldest-newest year order, but grouped by gene in Table 2).

  1. Table 3 (and all other tables), please include abbreviations in the end (also mention what “K” stands for (lysine residue) in the table.

All abbreviations were moved to below each table, and lysine (K) was added.

  1. Please include a statement in section 5 (Non-coding RNAs) that the review focuses mainly on data collected on miRNA and lncRNA.

Last sentence of first paragraph now reads “In PD, the most well-studied of these are miRNAs and this section will focus on miRNA and lncRNA expression studies”

8.      Very minor language edits required – for example, consecutive paragraphs (lines 539 and 549) start with “in addition” (which does not read well). Suggested title change for consideration - “An overview of epigenetic changes in the Parkinson disease brain”

Removed repeated “In addition”, changed title to above but with “Parkinson’s” instead of “Parkinson”

9.      Section 6.3, all points needn’t start with a separate line/para, please condense it all and make it continuous.

Future directions section is now one continuous paragraph. Note this is section 6.4 - in the version sent back to us the heading for section 6.2 “Utility of combining genetics and epigenetics studies” was removed as a heading.

In the revised version, we have corrected this with section 6.2 “Utility of combining genetics and epigenetics studies”, section 6.3 “Epigenetic therapies” and 6.4 “Future directions”.

All in all, a detailed and very well-organized review paper.

Comments on the Quality of English Language

Very minor edits needed. 

Reviewer 2 Report

Comments and Suggestions for Authors

Reviewer suggestions

The present article discusses the role of epigenetic changes in Parkinson’s disease (PD). PD is one of the most common neurodegenerative disorders affecting approximately more than ten million individuals worldwide. Epigenetic changes including histone modification, DNA methylation, lncRNA, and miRNA upregulation or downregulation lead to PD.

The article has discussed previously published work showing the potential role of epigenetic changes in PD in detail.

Recommendation

1.      The article is well-planned and written very well.

2.      The article has up-to-date references with sufficient numbers.

3.      Authors have discussed every aspect of epigenetics associated with PD.

4.      The article is free from typo errors and has sufficient tables.

Scientific comments

1.      It's better to add one figure showing the potential role of epigenetic changes in PD.

2.      Authors can add one figure showing the potential mechanism of action of miRNA/lncRNA in the pathophysiology of PD.

3.      Table 4, provides miRNA-associated pathophysiology in PD i.e. miRNA targets and their role in causing PD.

4.      In Table 5, add one more column for the mechanism of lncRNAs in PD or associated pathophysiology in PD.

Author Response

We thank the Reviewer for taking their time to revise our manuscript and for their useful comments. Please find our answers to the Reviewer's comments below.

REVIEWER 2:
Reviewer suggestions

The present article discusses the role of epigenetic changes in Parkinson’s disease (PD). PD is one of the most common neurodegenerative disorders affecting approximately more than ten million individuals worldwide. Epigenetic changes including histone modification, DNA methylation, lncRNA, and miRNA upregulation or downregulation lead to PD.

The article has discussed previously published work showing the potential role of epigenetic changes in PD in detail.

Recommendation

1.      The article is well-planned and written very well.

2.      The article has up-to-date references with sufficient numbers.

3.      Authors have discussed every aspect of epigenetics associated with PD.

4.      The article is free from typo errors and has sufficient tables.

Scientific comments

1.      It's better to add one figure showing the potential role of epigenetic changes in PD.

Figure 6 was added to summarise key findings related to DNA methylation, histone modifications and miRNAs in PD.

2.      Authors can add one figure showing the potential mechanism of action of miRNA/lncRNA in the pathophysiology of PD.

MiRNAs are included in this new Figure 6.

3.      Table 4, provide miRNA-associated pathophysiology in PD i.e. miRNA targets and their role in causing PD.

PD associated pathophysiology column was added to the table, listing key targets and/or effect of miRNAs on disease, for studies where this is reported.

  1. In Table 5, add one more column for the mechanism of lncRNAs in PD or associated pathophysiology in PD.

PD associated pathophysiology column was added to the table, listing key targets and/or effect of lncRNAs on disease, for studies where this is reported.